# Clinical Validation of a Colorimetric Loop-Mediated Isothermal Amplification Using a Portable Device for the Rapid Detection of SARS-CoV-2

**DOI:** 10.3390/diagnostics13071355

**Published:** 2023-04-06

**Authors:** Bruna W. Raddatz, Felipe J. Rabello, Rafael Benedetti, Gisleine J. Steil, Louise M. Imamura, Edson Y. S. Kim, Erika B. Santiago, Luís F. Hartmann, João V. Predebon, Bruna M. Delfino, Meri B. Nogueira, Jucélia S. dos Santos, Breno G. da Silva, Diego R. P. Nicollete, Bernardo M. M. de Almeida, Sergio R. Rogal, Marcus V. M. Figueredo

**Affiliations:** 1Hilab, Rua José Altair Possebom, 800-CIC, Curitiba 81270-185, PR, Brazil; 2Virology Laboratory, Universidade Federal do Paraná (Hospital de Clínicas), Rua General Carneiro, 181-Alto da Glória, Curitiba 80060-900, PR, Brazil

**Keywords:** RT-LAMP, colorimetric, SARS-CoV-2, Point-of-Care, diagnostics

## Abstract

Quick and reliable mass testing of infected people is an effective tool for the contingency of SARS-CoV-2. During the COVID-19 pandemic, Point-of-Care (POC) tests using Loop-Mediated Isothermal Amplification (LAMP) arose as a useful diagnostic tool. LAMP tests are a robust and fast alternative to Polymerase Chain Reaction (PCR), and their isothermal property allows easy incorporation into POC platforms. The main drawback of using colorimetric LAMP is the reported short-term stability of the pre-mixed reagents, as well as the relatively high rate of false-positive results. Also, low-magnitude amplification can produce a subtle color change, making it difficult to discern a positive reaction. This paper presents Hilab Molecular, a portable device that uses the Internet of Things and Artificial Intelligence to pre-analyze colorimetric data. In addition, we established manufacturing procedures to increase the stability of colorimetric RT-LAMP tests. We show that ready-to-use reactions can be stored for up to 120 days at −20 °C. Furthermore, we validated both the Hilab Molecular device and the Hilab RT-LAMP test for SARS-CoV-2 using 581 patient samples without any purification steps. We achieved a sensitivity of 92.93% and specificity of 99.42% (samples with CT ≤ 30) when compared to RT-qPCR.

## 1. Introduction

The 2020 SARS-CoV-2 pandemic highlighted the critical importance of molecular diagnostics as a tool to identify infected people and control the dissemination of the pathogen [1]. Reverse transcription followed by quantitative polymerase chain reaction (RT-qPCR) is the preferred method of choice for diagnosing several infectious diseases, and it has been widely used to detect the presence of SARS-CoV-2. However, its use has been compromised by a global shortage of reagents, supplies, and laboratory resources caused by the pandemic [1,2,3,4,5]. The technique requires highly specialized personnel, complex equipment, and an extended turnaround time for test results. Taken together, those constraints may limit and jeopardize a crucial part of contact tracing and isolation [6,7].

Alternative decentralized Point-of-Care (POC) testing platforms may greatly improve testing range and provide fast and reliable diagnostics for physicians and patients in less than an hour [6,8]. The detection of SARS-CoV-2 nucleic acids using amplification techniques only requires specific primers; meanwhile, antigen detection depends on the production speed of diagnostic antibodies [1]. Loop-mediated isothermal amplification (LAMP) based tests can amplify the target nucleic acid sequences under isothermal conditions in 30 min [9,10]. The reaction requires four to six primers and produces concatemers of double-stranded amplification products [9,11]. Strategies for amplification detection vary but are either direct—using DNA intercalating fluorophores for instance [9]—or indirect [12], including the colorimetric detection by a pH indicator dye [13].

The pH-dependent colorimetric detection allows the result to be read out by the naked eye since this technique produces a significant amount of DNA (up to 10^10^ copies, in contrast to PCR, which produces about 10^6^ copies after amplification) [14]. In the colorimetric reaction, whenever the DNA polymerase synthesizes a new DNA strand, it releases pyrophosphate and protons as a byproduct. The protons released decrease the pH of the reaction, changing the color from pink to yellow when using a pH-sensitive dye such as phenol red, which is especially important in resource-limited settings. However, human color perception is greatly affected by subjectivity, influenced by genetics, and the unique experiences of each individual [15]. Furthermore, the readout of intermediary results becomes difficult, so low viral load samples can be misinterpreted [16].

Therefore, in order to improve the assertiveness of the diagnostic—avoiding misinterpretation due to the subjectivity of color readout by the naked eye we describe here the clinical validation of a RT-LAMP test for SARS-CoV-2 and a POC system for molecular diagnosis. The clinical validation was performed with 581 patient samples using the Hilab Molecular POC device, which is a real-time and robust system based on the Internet of Things (IoT) and Artificial Intelligence (AI). We also establish manufacturing procedures that maintain the stability of ready-to-use RT-LAMP tests above 120 days, while reducing the rate of false-positive results.

## 2. Materials and Methods

### 2.1. Primers and Positive Control

The primer sequences used in this study, previously described elsewhere [17], targeted the genes N (nucleocapsid) and E (envelope) from the SARS-CoV-2 genome and human rActin gene for internal control (Appendix A). Primers were resuspended in ultrapure nuclease-free water (Invitrogen, Waltham, MA USA, #10977015), fractionated into aliquots, and stored at −20 °C. Aliquots were thawed as minimally as possible to avoid oligonucleotide degradation. Primer mix was prepared as a duplex 10× for SARS-CoV-2 gene N and E (16 µM FIP/BIP, 2 µM F3/B3, 4 µM LoopF/B of primer of each target), and as a singleplex 10× mix targeting human rActin (16 µM FIP/BIP, 2 µM F3/B3, 4 µM LoopF/B).

Positive controls using gene N and E sequences were synthesized by GenScript (Appendix A), sequenced, and cloned into pUC17 in the EcoRV site (pUC17_N and pUC17_E). To quantify the control plasmid copy number, primers Fw_pUC57_qPCR2 and Rev_pUC57_qPCR2 (Appendix A) were used in a quantitative real-time PCR assay. The plasmid pUC57 (Thermo Scientific™, Waltham, MA USA, SD0171) was used to obtain a standard curve from serial dilutions. The assay targeted the *bla* gene from the plasmid and was also used to quantify both plasmidial positive controls by the comparison of CT (Cycle Threshold) values with the standard curve. For this assay, PowerUp SYBR Green Master Mix (Thermo Fisher) was used, along with a Chai Open qPCR real-time thermocycler. A control mixture containing each pUC17_N and pUC17_E at a concentration of 10^6^ copies per µL was prepared in ultrapure nuclease-free water (Invitrogen, #10977015), aliquoted, and then frozen at −20 °C.

### 2.2. RT-LAMP Assays

In order to prepare the master mixes, all reagents were thawed, briefly homogenized with vortex, and spun down before pipetting. Reagents were kept in an ice bath and were only opened inside a biological safety cabinet. Assays were assembled in a total reaction volume of 12.5 µL. Each reaction contained 6.25 µL of the WarmStart Colorimetric RT-LAMP 2× Master Mix (New England Biolabs, Ipswich, MA, USA, M1800), 1.25 µL of the 10× primer mix, and 1.25 µL of guanidine hydrochloride 400 mM (Sigma-Aldrich, Saint Louis, MO, USA, G3272). Internal control and test tubes had the addition of 2 µL of HS (hot swab—nasopharyngeal swab samples after heat inactivation at 95 °C for 10 min). In the positive control tube, 1 µL of each plasmid (pUC17_N and pUC17_E) at 10^6^ copies per µL was added. In the negative control tube, only 2 µL of ultrapure type I nuclease-free water (Invitrogen, #10977015) was added. The remaining volume for a final 12.5 µL reaction was filled with ultrapure type I nuclease-free water (Invitrogen, #10977015). When the amplification was detected by fluorescence, 0.625 µL of ChaiGreen 20× (Chai BioTechnologies, Santa Clara, CA, USA, #R01200S) was included in the reaction and the corresponding volume was removed from the water fraction. To minimize the acidification of reagents due to the exposure to the atmospheric CO_2_, the tubes were kept open as minimally as possible and the colorimetric reagent was only added to the reaction as a final step. Additionally, RT-LAMP reactions were assembled in microtubes suitable to the total volume being prepared, avoiding excessive headspace in the tubes. Master mixes were prepared in small batches (maximum of 192 reactions) and all the steps were conducted on an ice surface or in an iced bath. For the aliquoting, an electronic pipette with a multi-dispensing function was used for time optimization and to reduce the contact of the reaction with air. The reactions were immediately used in experiments or frozen at −20 °C for up to 120 days.

Reactions were performed on the Hilab Molecular device (Hilab) or the Open qPCR (Chai Biotechnologies) single-channel thermocycler, by incubating at 65 °C for 30 min. In colorimetric reactions, to improve color reading by the equipment, the temperature was lowered to 40 °C for 1 min [1].

### 2.3. Limit of Detection (LoD)

To evaluate the lowest number of copies of the target sequence that produces a positive result, we performed a serial dilution of the plasmid pUC57 containing the sequences of genes N or E from the SARS-CoV-2 genome (Appendix A). A serial dilution of the plasmids was used to achieve a log curve ranging from 0 copies/reaction to 10^6^ copies/reaction of each gene. Amplification was evidenced by color change detected by the Hilab device. Each titration point was performed in 10 technical replicates. LoD was defined as the dilution at which 100% of the replicates correctly amplified the initial target sequences.

### 2.4. Reproducibility

To verify the reproducibility of the test, a serial dilution of the positive controls (pUC17_E and pUC17_N) was made to a final concentration of 10^3^, 10^4^, and 10^5^ of each control, per reaction. The analysis was performed on three different days, and each day was executed by a different operator. Sufficient tests were produced each day to mimic three different batches. Each run consisted of ten technical replicates, with positive and negative controls (Appendix A).

### 2.5. Clinical Validation

Samples of healthcare workers from a tertiary public hospital in Curitiba, Brazil, at the Hospital de Clínicas/Universidade Federal do Paraná, were collected and analyzed by the Virology Laboratory of the same institution. Samples were collected in two moments, from July to November 2021, and January 2022. A total of 581 volunteers participated by donating two nasopharyngeal (NP) swabs, one to be used for RT-qPCR and the other for RT-LAMP. Samples in 2021 for RT-qPCR were collected in a viral transport medium (VTM), while in 2022 they were collected in an antigen-test buffer (possibly containing salts, surfactants, proteins, preservatives, and blockers; the precise composition of the buffer is unknown, and proprietary). Samples for RT-LAMP were collected in the sample solution provided with our test (10% TE buffer pH 8.0, 0.1% Tween 20, and ultrapure water to a final volume of 500 µL).

The tests were performed by colorimetric RT-LAMP in the Hilab Molecular device and by RT-qPCR in the 7500 Real-Time PCR Applied Biosystems™. In the Hilab kit, each run was composed of a negative control tube (genes N and E of SARS-CoV-2 as the target; addition of ultrapure type I nuclease-free water instead of a sample), a positive control tube (genes N and E of SARS-CoV-2 as the target; addition of plasmid pUC17_N and pUC17_E at 10^6^ copies per µL, also with no sample addition), an internal control tube (human rActin as the target; addition of sample after heat inactivation), and the test tube (genes N and E of SARS-CoV-2 as the target; addition of sample after heat inactivation). The swabs collected for the RT-LAMP assay were homogenized in the sample solution. After a heat inactivation (95 °C for 10 min), 2 µL of the sample was added to the RT-LAMP mix, as described previously.

The swabs collected for the RT-qPCR assay were homogenized in 1000 µL of VTM or antigen-test buffer, and 100 µL of each sample was extracted using automated extraction (Extracta Kit FAST—Loccus) to obtain the viral RNA. The RNA was added in a volume of 5 µL to the Kit BIOMOL OneStep/COVID-19 (IBMP) RT-qPCR mix. The RT-qPCR targeted the N and ORF1ab genes of SARS-CoV-2. Eleven samples were collected on the weekend or were emergency cases; therefore, for these samples, the analyses were made with GeneXpert Instrument Systems (Cepheid) that targeted the E and N genes of SARS-CoV-2 (Appendix A). Three other samples were analyzed targeting the N1 and N2 genes of SARS-CoV-2 with in-house technology by the biochemical department of Universidade Federal do Paraná when the Virology Laboratory was lacked of resources or in high demand. Discordant samples with negative RT-PCR results and positive RT-LAMP results were retested from the purified RT-LAMP product with the BIOMOL OneStep/COVID-19 Kit (IBMP) RT-qPCR mix to verify if the amplification was specific.

This study was approved by the local Ethics Committee (CAAE 31687620.2.0000.0096). The samples were obtained under the signature of the Free and Informed Consent Form from July 2021 to January 2022.

### 2.6. Cross-Reactivity

Lyophilized controls containing the inactivated viruses of Influenza A (Controllab, Rio de Janeiro, RJ, BR ATG–129), Influenza B (Controllab, ATG–103), and Respiratory Syncytial Virus (RSV) (Controllab, ATG–122) were reconstituted using ultrapure nuclease-free water according to the manufacturer instructions. The controls were incubated at 95 °C for 10 min (same protocol as HS) and only then were pipetted into colorimetric LAMP reactions with SARS-CoV-2 primers to verify a possible cross-reaction.

### 2.7. Long-Term Stability at −20 °C

Aiming to determine how long a ready-to-use colorimetric RT-LAMP reaction could be stored when frozen at −20 °C, we performed a real-time stability test. Colorimetric LAMP tests for SARS-CoV-2 were prepared as previously described, aliquoted in 0.2 mL nuclease-free microtubes, assembled in the Hilab Molecular cartridge, sealed into an aluminized zip lock bag and frozen at −20 °C. Every 30 days, until day 120, a duplicate was taken out, thawed, and tested. The tests consisted of a positive control tube, containing SARS-CoV-2 genes E and N each at 10^6^ copies per reaction, and a negative control test tube with primers but with no template DNA. Tests were performed using freshly collected NP swabs after heat inactivation. Results were obtained after 30 min at 65 °C, followed by a cooling step to 40 °C.

### 2.8. Hilab Molecular Device

The data collection (RGB codes) was done in real-time and the information was sent by cloud for analyses by biomedical professionals. After the collection for 30 min, the information was converted into two images to assist the analyses plus the patient anamnesis questionnaire. The first image consists of the color change of the tube reactions during the 30 min, and the second image is a curve with the delta of the color change. The result issued to the patient is a qualitative result (positive or negative). However, for internal information, it is also possible to analyze the initial time of an amplification (Cy0) [18].

## 3. Results

### 3.1. Hilab Molecular Device Overview

The Hilab Molecular device is a novel portable nucleic acid detection system, part of an integrated diagnostic platform that uses the Internet of Things (IoT) to deliver high-quality data to expert healthcare professionals, which generates an AI-assisted diagnostic (Figure 1a). The device dimensions are 8.8 cm × 8.6 cm × 9.9 cm (width × depth × height) and it weighs 0.22 kg, so it can be easily transported to different test locations. The portable device has five wells—one is for the sample inactivation (1.5 mL tube) and the other four are for the RT-LAMP reaction tubes (0.2 mL tubes) (Figure 1b).

The workflow of the molecular test is simple and can be easily done by health professionals in a few steps. A health professional collects a nasopharyngeal (NP) swab sample and then swirls it in a microtube containing the sample solution. After homogenization, the swab is discarded and the closed microtube is placed into the Hilab device to undergo heat inactivation. After inactivation, using a capillary pipette, the sample is pipetted into two reaction tubes, the internal control tube—to ensure the nasopharyngeal sample was really collected—and the test tube. Each cartridge also has negative and positive controls for SARS-CoV-2 to certify that the reaction is working correctly. Then, the test cartridge is inserted into the Hilab device to amplify the target DNA/RNA.

When the test is completed, the amplification data is directed to a healthcare professional (Figure 2), who will issue the signed report while being assisted by artificial intelligence. The whole process takes approximately 1 h and the result is sent to the patient by SMS, e-mail, or the Hilab App.

### 3.2. Hilab Molecular Device Principles of Operation

The main sensor/actuator assembly consists of a custom machined aluminum block and a 3D printed heater cartridge, located at the center and cooled by a fan at the bottom. The temperature of the block can be precisely controlled to ±0.2 °C with a range of 40–98 °C, acquiring it using a factory-calibrated negative temperature coefficient sensor (NTC), mounted at the same distance as the sample wells, to compensate for reading delays due to heat capacity in the material.

There are five wells in the device, which are distributed radially to the heater in order to achieve an equal distribution of heat in every sample. Figure 3 represents the detection system for each cavity.

To detect nucleic acid amplification, the device detects the transluminance of the sample using a white LED, mounted on the top surface of the block, and a luminosity sensor with mounted filters on the bottom surface. To maintain the inner sensor board components under optimal operating temperatures, the fan is activated when a 70 °C temperature is measured on the Printed Circuit Board NTC.

The device also has a safety system to prevent possible failures in the temperature controller system. When an abnormal resistance is measured for one of the NTCs or a thermal decoupling between the sensor and heater occurs, the controller software shuts down and the device enters temperature fail mode (TFM).

When in TFM, the device disables the heating, as well as sets the fan to its maximum power. The system automatically notifies both the user and technical support that a recall is needed. If all digital safety measures fail, there is an analog protection circuit, which is able to engage TFM automatically when a temperature over 110 °C is detected.

### 3.3. Limit of Detection (LoD)

We considered the LoD to be the dilution in which 100% out of 10 replicates produced correctly positive results (Table 1, Appendix A). The limit of detection found was approximately 156 copies of each gene per µL (1.95 × 10^3^ copies in a 12.5 µL reaction).

### 3.4. Reproducibility

We found 100% of reproducibility in the three different levels of target tested using different batches and performed by different operators (Appendix A).

### 3.5. Clinical Validation

Table 2 summarizes the results obtained from the clinical validation using NP swab samples (raw data is available in the Appendix A). Compared to the gold standard RT-qPCR technique, the RT-LAMP test showed a sensitivity of 84.44% (79–89%; Confidence Interval (CI): 95%), specificity of 99.42% (98–100%; CI 95%), and an Overall Percentage Agreement (OPA) of 93.53% (Table 2). Figure 4 demonstrates the comparison between RT-qPCR and RT-LAMP results. For the RT-LAMP assay, a positive sample is determined based on the initial time of amplification (Cy0)—which has to be <1200 s—and a delta, the color change of the reaction, above 15. These criteria are linked to the anamnesis questionnaire to achieve the final result.

When we categorized the samples by the CT (cycle threshold) obtained from RT-qPCR (Table 3), considering only the positive samples with CT ≤ 30 (92) the test presented a sensitivity of 92.93% (86–97%; CI 95%), specificity of 99.44% (98–100%; CI 95%) and OPA of 98.02%.

### 3.6. Cross-Reactivity

To verify the possible cross-reaction with other respiratory viruses, we used the SARS-CoV-2 colorimetric RT-LAMP test with commercially available molecular controls for Influenza A, Influenza B, and Respiratory Syncytial Virus (RSV). Results indicated no cross-reactivity of the primers used (Appendix A) with the tested controls (Figure 5).

### 3.7. Long-Term Stability at −20 °C

Our results suggest that ready-to-use reactions are stable with no unspecific amplification in negative controls, and positive controls changed their color to yellow as expected (Figure 6).

### 3.8. Manufacturing Procedures to Ensure Long-Time Stability

Since colorimetric RT-LAMP reactions tend to become yellowish when manipulated [1], we ought to understand which procedures could be done when mass-producing it to maintain its original pink color as long as possible.

Considering that colorimetric RT-LAMP master mixes are low-strength buffered solutions, the color change could be attributed to the acidification by atmospheric CO_2_. To minimize this, we established a few precautions that should be taken when producing the ready-to-use tests. 


Keep all reagents—even the thermostable reagents—in an iced bath. The higher the temperature of the reagents, the faster the incorporation of atmospheric CO_2_ is;Use Safe-Lock microtubes to reduce gas exchange between the atmosphere and the headspace in tubes;Keep tubes closed when not pipetting;Add Master mix as the last reagent in the reaction;Use higher volumes of reactions, as this will reduce the headspace in tubes and increase the total amount of CO_2_ that can be incorporated without changing the reaction color;When aliquoting the reaction volume, preferably use reverse pipetting since it does not add air to the solution;Avoid homogenizing the reaction by vortex, instead, preferably pipette mix the reagents;


Store the tubes in sealed packs with a molecular sieve (4A) as a desiccant, since it is capable of absorbing both humidity and CO_2_.

## 4. Discussion

The interplay of several factors, more often than not, influences the outcome of how and how fast a new pathogen is going to spread. Being so, agile, real-time, and integrated diagnostics provide crucial epidemiological information and allow decision-makers to take data-driven measures such as lockdowns and mobility restrictions [19]. Within this manuscript, we described the clinical validation of a molecular test to diagnose COVID-19 using the Hilab Molecular system, an RT-LAMP-based POC device, developed to provide diagnostic results directly to the hands of the patient and/or physician in less than 1 h.

Although colorimetric RT-LAMP assays can be performed in equipment as simple as a thermoblock, cheap instruments usually have poor temperature control, which can influence the formation of secondary structures and primer dimers leading to false-positive results [20]. The same can be said of the result read out by the naked eye. Although it facilitates the identification of positive results, low viral load reactions can be often misinterpreted [15,16,21]. The Hilab Molecular device described here solves both of these issues. It features a precise temperature controller and a colorimetric sensor, and since it is integrated with the Internet of Things, color and temperature data can be transmitted in real-time to the laboratory headquarters. This adds more robustness to the test since the equipment can quantify the color change of the reaction and give the initial time of amplification, avoiding misinterpretation and consequently false-positive results.

Compared to other developments, our device uses a simpler data treatment while maintaining its robustness. Problems with bubbles or lightning exposure viewed in another study [22] have been bypassed by using a color sensor at the bottom of the microtubes, as well as instructing the operator to perform a manual spin to clear any remaining bubbles in the solution. In addition, for each color capture (every 10 s) a standardized white light is emitted, guaranteeing that the same color pattern was captured. This is a clear advantage over smartphone capture, since the lightning exposure and equipment brand has a strong influence on the data acquisition [23]. Another advantage of our device is that it captures the data automatically. Otherwise described elsewhere [24], the operator does not have to monitor or record the whole amplification reaction. In addition, the data treatment is also automatized; therefore, the health professional only evaluates the color acquisition in parallel with the anamnesis questionnaire. The operator and the patient receive the diagnostic by SMS or email, and no additional action is required. Additionally, the device is small and can be easily carried to remote settings, decentralizing molecular testing, which is especially important in pandemic scenarios. While other molecular tests still rely on equipment such as microplates reader [25]. However, a limitation of the Hilab Molecular is the sample number that can be analyzed at a time; only one sample plus the positive, negative, and intern control. Another device, with similar technology and space for eight microtubes, has been described elsewhere [26].

Here, we found that our colorimetric RT-LAMP test has an LoD of approximately 156 copies of each gene per µL, which is compatible with other isothermal POC without RNA extraction [27]. Sample purification can be used in order to achieve a lower LoD, however, this could hinder its use as a POC test. Nonetheless, the achieved LoD is compatible with the viral load of SARS-CoV-2 found in nasopharyngeal swabs samples [28]. This finding acknowledges the use of this colorimetric RT-LAMP test as an efficient POC tool for the detection of SARS-CoV-2 infection. 

In the clinical validation, the NP swab samples for RT-LAMP were collected in our own developed sample solution, while the samples used in RT-qPCR were collected in either Viral Transport Medium (VTM) (2021) or an antigen-test buffer (2022). It should be taken into account that our assay uses non-purified samples, while the RT-qPCR test used purified RNA from samples. The RT-LAMP test had an overall agreement with the RT-qPCR gold standard of 98.02% for samples with CT below 30, with a sensitivity of 92.93% (86–97%; CI 95%) and specificity of 99.44% (98–100%; CI 95%). Those results are similar to other RT-LAMP tests using purified samples [24]. Considering all samples (including with CT > 30) our OPA is 93.6% compared to RT-qPCR, higher than another study that uses purified samples [25]. 

Low viral load samples (CT > 30) had lower sensitivity (77.78–70%-84; CI 95%), which can be attributed to the fact that samples for RT-LAMP were more diluted and were not purified as was the case for RT-qPCR. Taking this result together with the limit of detection of 1.95 × 10^3^ copies per reaction obtained for the test, we can infer that samples with CT above 30 should have less than 1.95 × 10^3^ copies of the virus in the reaction (which represents approximately 10^6^ copies/mL of the sample). However, it should be noted that the Cy0 obtained from LAMP does not correlate properly with RT-qPCR CT or viral load, since we saw that high viral load samples usually have similar Cy0. This is due to the fact that LAMP reactions have complex amplification kinetics and colorimetric reactions need a minimal number of amplified products for the pH-dependent dye to change its color. Previous work evaluated the viral load of SARS-CoV-2 in nasopharyngeal swab samples of people with variable disease severity. The average viral load found was above 10⁷ copies/mL, with no statistical difference regarding disease severity [29]. Another study evaluated the viral load in self-collected nasal swab samples. Although a few samples (6/43) had a viral load below 10⁵copies/mL, most of them were above 10^7^ copies/mL [30]. Thus, considering that the Hilab Molecular test was able to detect 4.88 × 10^2^ copies (representing 2.5 × 10⁵ copies/mL of the sample) in 80% of the cases, it still would be able to identify a majority of the infected people, even in relatively low viral load individuals. 

The choice to use the antigen-test solution in 2022 for RT-qPCR was caused by the emergence of the Omicron variant and the second wave of COVID-19 cases. The number of cases rose quickly, and antigen tests were more efficient in detecting individuals with active viral replication [31,32]. In addition, this system would avoid the need for collecting three nasopharyngeal samples from the same person, considering that one sample was already used for RT-LAMP. We observed that RT-qPCR tests with samples collected in the antigen-test solution had an increase in the overall CT when compared to the samples in VTM (Appendix A). We hypothesize that the solution either interfered with the RT-qPCR or could not maintain the stability of the SARS-CoV-2 RNA [33]. 

Furthermore, we also evidenced an increase in the initial amplification time (Cy0) in the RT-LAMP assays with samples collected in 2022. While the solution composition remained the same throughout the whole validation, we believe this is due to a mutation in the E gene of the Omicron variant (T9I), the same region targeted by our FIP primer (Appendix A). Therefore, it would be reasonable that the Cy0 increased, since this mutation would hinder most amplification of the E gene, and the only product would be due to the amplification of the N gene. However, confirmation of this hypothesis is only possible with the sequencing of the samples, to verify the SARS-CoV-2 variant. Besides this, if new COVID-19 variants added mutations at the N gene, specifically in the region of our primer annealing, it is possible that our test would not be able to detect these variants. Nonetheless, it is unlikely that a new variant has a mutation in the N region since it is a highly conserved region [34].

Since SARS-CoV-2 and other respiratory viruses have very similar clinical manifestations but different treatments, it is important to verify whether the designed primers would be specific for SARS-CoV-2 detection. Using commercially available controls for Influenza-A, Influenza-B, and RSV, we found no indication of cross-reactivity with the primer set designed for SARS-CoV-2. Although more comprehensive studies of current circulating viruses are required, this indicates that the primer set is suited for specific detection of SARS-CoV-2 infection.

Previous reports indicated that colorimetric RT-LAMP master mixes, with primers included, are not exceptionally stable. According to Dao Thi et al. (2020) [1], storage at −20 °C for more than 18 h may increase the rate of non-specific amplification in a 30-min reaction. Despite those reports, in this work, we were able to achieve highly stable LAMP reactions up to 120 days after being frozen at −20 °C, showing no color change in the negative controls even when compared with freshly prepared reactions. Up to the last day tested, all positive reactions had a color change from pink to yellow, as expected. To be able to mass-produce colorimetric RT-LAMP tests while maintaining their color quality, we established a set of procedures to be followed. Maintaining the laboratory temperature at 18 °C, always keeping the reagents in an ice bath, adding the master mix containing phenol red at the last production step, closing the tubes right after aliquoting, and using an electronic pipette had a notable improvement in the stability of the test. We observed that if these steps weren’t followed, the RT-LAMP reaction color rapidly turned from pink to orange, indicating acidification of the solution due to the incorporation of atmospheric CO_2_. Another study that performed some of these procedures also saw an improvement in the stability of the color reaction [23].

## 5. Conclusions

The Hilab Molecular colorimetric RT-LAMP POC test to diagnose COVID-19 was validated with 581 nasopharyngeal samples, without any step of purification, with a sensitivity of 92.93%, specificity of 99.44% and OPA of 98.02% for samples with a CT < 30. However, our study has some limitations, such as the number of positive samples, which was only 33%. A higher number of positive samples should be tested to improve our clinical validation. The test presented an LoD compatible with POC applications and thus is an important resource to be explored as a complementary methodology in epidemiological contingency. A set of practices were established to avoid acidification of the RT-LAMP reaction during preparation, and adequate stability up to 120 days at −20 °C was accomplished. The primer set used in this study exhibited no amplification when combined, in colorimetric LAMP reactions, with control samples containing Influenza-A, Influenza-B, and RSV. More experiments must be performed to analyze if the LoD and the specificity of the test remain unaltered after 120 days.

## 6. Patents

This work resulted in three patents, of which two are patent drawings (BR 302020006117-9; BR 302020006116-0; BR 102021011067-8).

Additionally, our test has been registered in ANVISA, the Brazilian Health Regulatory Agency (80583710026).

## Figures and Tables

**Figure 1 diagnostics-13-01355-f001:**
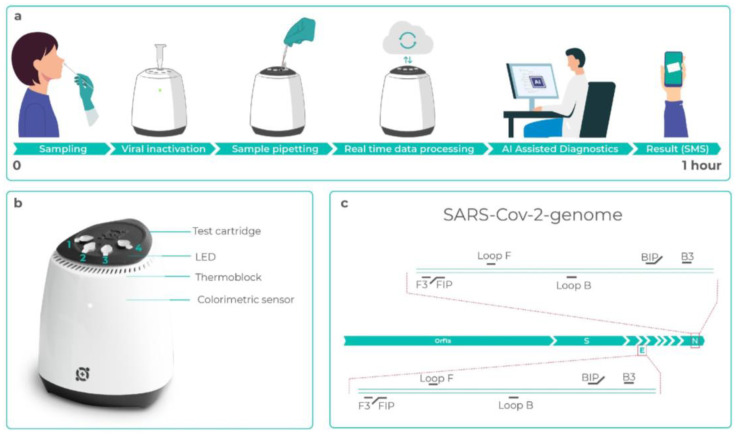
The Hilab Molecular system. (**a**) From swab collection to result in a timeline, describing the integrated system workflow. Samples are first submitted to a heat process for viral inactivation and then pipetted into the reaction tubes. The whole test can be monitored in real-time and the diagnosis is given by a healthcare professional assisted by artificial intelligence. Patients receive the results automatically by SMS or email in less than 1 h. (**b**) The molecular Hilab device. Each SARS-CoV-2 test cartridge is composed of negative control (1), internal control (2), test tube (3), and positive control (4). The colorimetric measurements are performed using an LED at the top and a colorimetric sensor under the tubes. (**c**) Schematic representation of the SARS-CoV-2 genome, with detailed information about the primers sites for genes E and N.

**Figure 2 diagnostics-13-01355-f002:**
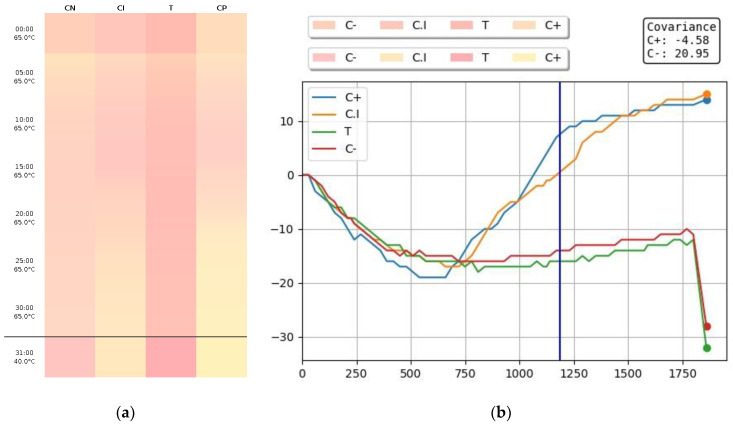
Colorimetric LAMP data. After 30 min of incubation, two images are generated: (**a**) is the first image which consists of the color change of the tube reactions during the 30 min, and the second image (**b**) is a curve with the delta of the color change. In graph (**b**), a different color represents each tube. The negative control tube is represented in red, the internal control tube in orange, the test tube in green, and the positive control in blue. The *X*-axis represents the time of the reaction in seconds. These images are sent for analysis by a healthcare professional with the anamnesis questionnaire.

**Figure 3 diagnostics-13-01355-f003:**
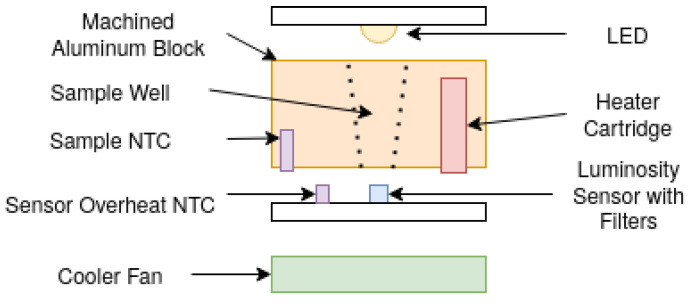
The Hilab Molecular sensor diagram.

**Figure 4 diagnostics-13-01355-f004:**
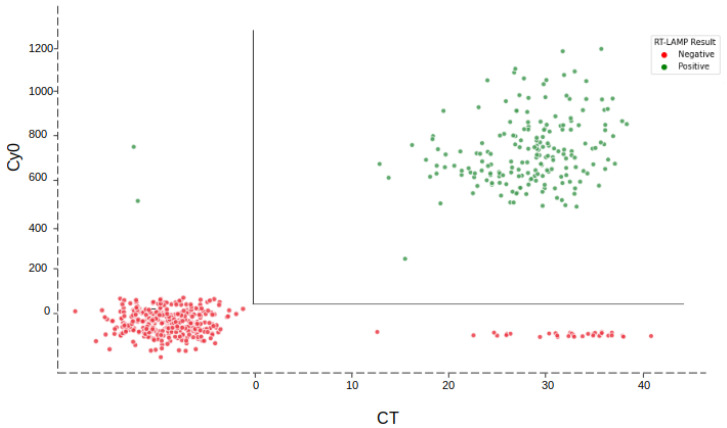
Comparison between the CT (cycle threshold) obtained from RT-qPCR for the gene N and the Cy0 (seconds) from RT-LAMP. The colors represent the qualitative result of RT-LAMP. Positive samples in RT-LAMP (Cy0 < 1200 s) are displayed as green dots and negative samples as red dots.

**Figure 5 diagnostics-13-01355-f005:**
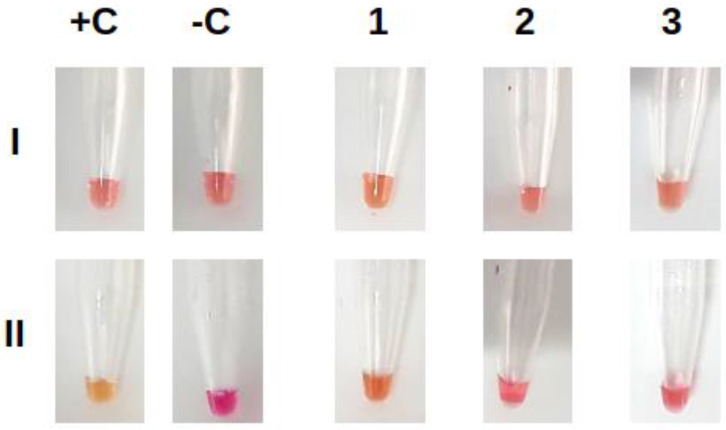
Cross-reaction experiment. SARS-CoV-2 primers were tested against Influenza A (1), Influenza B (2), and RSV (3). Images were captured before (I) and after (II) the reaction. Positive (+C) and negative (−C) controls are shown.

**Figure 6 diagnostics-13-01355-f006:**
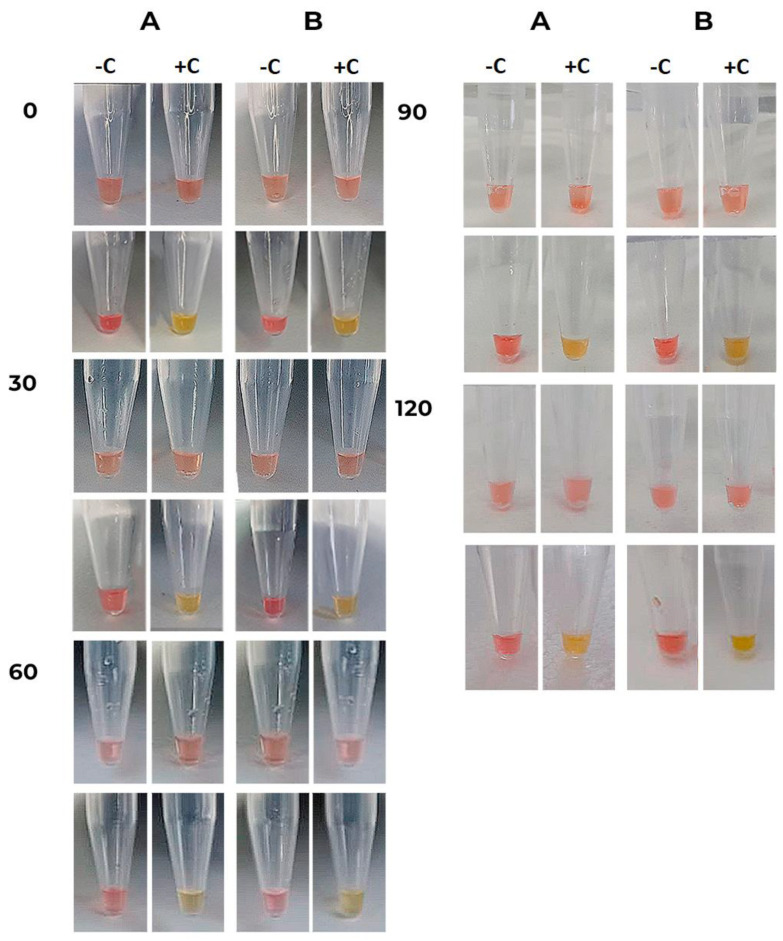
Colorimetric RT-LAMP long-term stability. In 30-day intervals, the stability of the reaction was tested in 2 replicates (A and B), with photos of the tubes right after the thawing of the reagents (Upper lane) and also at the end of the reaction (Lower lane). The corresponding day is given on the left side. The test was composed of a negative control (−C), with no DNA, and a positive control (+C) containing pUC57 plasmids cloned with the sequences for genes E and N of the SARS-CoV-2 genome in equal amounts.

**Table 1 diagnostics-13-01355-t001:** Colorimetric RT-LAMP Limit of Detection. Each dilution point contained equal amounts of pUC57 + gene E and pUC57 + gene N. 10 technical replicates were performed, and the number and the percentage of correct positive results are shown.

Target Copies per Reaction	Number of Replicates	Positive Results	% of Positive Results
1 × 10^6^	10	10	100%
1.25 × 10^5^	10	10	100%
1.5 × 10^4^	10	10	100%
**1.95 × 10^3^**	**10**	**10**	**100%**
9.77 × 10^2^	10	8	80%
4.88 × 10^2^	10	8	80%
2.44 × 10^2^	10	7	70%
1.22 × 10^2^	10	2	20%
0	10	0	0%

**Table 2 diagnostics-13-01355-t002:** Confusion Matrix of clinical validation.

Sample Results		RT-LAMP	
		Positive	Negative
RT-qPCR	Positive	190	35
	Negative	2	354
Sensitivity (95% CI)		84.44% (79–89%)	
Specificity (95% CI)		99.44% (98–100%)	
Overall Percentage Agreement (OPA)		93.63%	

**Table 3 diagnostics-13-01355-t003:** Validation parameters defined by the RT-qPCR CT.

CT	Sensitivity	Specificity	OPA
≤20	94.44%	99.44%	99.19%
20–25	91.67%	99.44%	99.18%
25–30	92.75%	99.44%	98.35%
>30	77.78%	99.44%	93.78%
≤30	92.93%	99.44%	98.02%

## Data Availability

The data that support the findings of this study are available in the Appendix A. Other data can be obtained from the corresponding author, B. R., upon request.

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
