# Peer review of "Clinical Validation of a Colorimetric Loop-Mediated Isothermal Amplification Using a Portable Device for the Rapid Detection of SARS-CoV-2"

_diagnostics, 2023, doi:10.3390/diagnostics13071355_

Round 1

Reviewer 1 Report

In fact, the detection of patients is very important to prevent the spread of infection. Antibodies can be used, but their sensitivity is too low; PCR is a good method, but it is laborious. In Japan, for example, even the exact number of patients remains unknown due to the continuing lack of it.

The LAMP method is the next best method and more convenient, but its disadvantage is that it does not have the clarity of qPCR, as it detects by suspension or colour change.

This paper compensates for this disadvantage and practical application is desirable. Naturally, this paper is also worthy of publication.

Some requests are listed below.

When the abbreviation CI is first used, it should be explained as the confidence interval.

Obviously, the most important figure is Fig. 4, which should discuss the low correlation between CT and Cy0 in qPCR. If Cy0 correlates with viral load, this may be a clue for treatment.

It is unfortunate that 10% of the cases are undetectable, even though there is fewer detection failure if it is below CT30. In particular, what does it mean that the detection of quite concentrated ones (about 13 on CT) is missed? I am very aware that this is not a paper on the development of the LAMP method, but are there any possible causes for this? Is it due to some contaminant? Or is it because of the short time taken to make a detection decision? In this regard, we would like to see a presentation like the one shown in Figure 2 in relation to, for example, 1-0.3E3 copies in Table 1. This is a highly important request.

Was the sample that was negative by qPCR but detected by LAMP, of which there were two, a false negative by qPCR? Maybe it has been found by sequencing the LAMP product. Despite being the gold standard, qPCR is not infallible. If you still have samples, why not give them a try?

Reviewer 2 Report

The authors present a colorimetric LAMP reaction combined with  Hilab Molecular device for SARS-CoV-2 detection that exploits artifical imteligence for interpretation of the results to give an answer if the analyzed sample is negative or positive. The authors have documented their work very well and clearly presented it.

However, the authors should clearly state the advantages of their method and device compared to all existing LAMP assays for SARS-CoV-2 e.g.:

Rohaim et al. Artificial Intelligence-Assisted Loop Mediated Isothermal Amplification (AI-LAMP) for Rapid Detection of SARS-CoV-2. Viruses 2020, 12, 972; doi:10.3390/v12090972

Avendo et al. A Rapid RT-LAMP Assay for SARS-CoV-2 with Colorimetric Detection Assisted by a Mobile Application. Diagnostics 2022, 12, 848. https://doi.org/10.3390/diagnostics12040848

Lim et al. Clinical validation of optimised RT‑LAMP for the diagnosis of SARS‑CoV‑2 infection. Scientifc Reports | (2021) 11:16193 | https://doi.org/10.1038/s41598-021-95607-1

Amaral et al. A molecular test based on RT‑LAMP for rapid, sensitive and inexpensive colorimetric detection of SARS‑CoV‑2 in clinical samples. Scientifc Reports | (2021) 11:16430 | https://doi.org/10.1038/s41598-021-95799-6

Lai et al. Colorimetric Reverse Transcription–Loop-Mediated Isothermal Amplification Assay for Rapid Detection of SARS-CoV-2. Am. J. Trop. Med. Hyg., 105(2), 2021, pp. 375–377

and also all other portable devices for LAMP reactions for SARS-CoV-2 detection such as:

Gangulia et al. Rapid isothermal amplification and portable detection system for SARS-CoV-2. PNAS | September 15, 2020 | vol. 117 | no. 37 | 22727–22735

Sreejith et al. A Portable Device for LAMP Based Detection of SARS-CoV-2. Micromachines 2021, 12, 1151. https://doi.org/10.3390/mi12101151

Diaz et al. Real-time optical analysis of a colorimetric LAMP assay for SARS-CoV-2 in saliva with a handheld instrument improves accuracy compared with endpoint assessment. Journal of Biomolecular Techniques 32:158–171 © 2021 ABRF.

Finally, reproducibility data should be included.

Reviewer 3 Report

Congratulation for your work and your efforts to support with your data the significance of your novel technique in the detection of SARS-CoV-2. We should always encourage investigators who orchestrate efforts and ideas to provide the scientific society with new and significant information.

Noteworthy the extensive analysis of your data provided valuable information but however there are some points that I would like to comment, and your response will be appreciated.

Comment 1

The number of the studied specimen was taken from 581 patients but the positive swaps were less than 200 which is a very small sample size to evaluate the true calculation of methods’ sensitivity. This has to be referred in the study limitation.

Comment 2

A method must be investigated for reproducibility. Please comment or submit your data.

Comment 3

We do know that the S part of the virus genome exerts its potential in the host cell contamination and all virus variants are categorized according to Spike protein mutation. While your technique does not recognize the spike protein virus gene, is there any possibility to be incapable for detection some virus variants? Please commend.

Comment 4

Please describe efficiently the details of the artificial intelligence tool you have incorporated in your system.

Comment 5

I believe that some parts of your manuscript presented in the study result should be better transferred in the methodology section.

Reviewer 4 Report

Raddatz and colleagues report on a SARS-CoV-2 LAMP assay which they evaluated. The evaluation of such a diagnostic tool fits into the scope of the journal. However, before publication is considered, I still have some recommendations on how to further improve the work.

1. Abstract: Considering the fact that the most parts of the world consider SARS-CoV-2 as endemic in the meantime, the abstract’s first sentence (calling for isolation and containment procedures) seems a little bit outdated. The authors might want to reconsider it, because rapid and reliable SARS-CoV-2-diagnostics do not require subsequent isolation procedures to be desirable but are of importance for individual patient care by themselves.

2. Methods: Considering the fact that the authors describe an in-house approach, do they plan any licensing of their product? As they might know, several countries (like, e.g., within the European union) demand the use of certified tests in line with the in-vitro diagnostics regulation for medical diagnostic approaches.

3. In the results chapter, there is an interesting contradiction. As defined with the dilution series of the positive control plasmids (table 1), the limit of detection (defined by 10 out of 10 correct identifications) is 156 copies per microliter. In contrast as shown in table 3, only 77.78% sensitivity was recorded for cases with Ct-values >30 in real-time PCR. Although not explicitly stated by the authors, a Ct-value of 30 usually translates into 1,000,000 copies. According to table three, sensitivity >90% is only to be expected in case of Ct </= 30 and thus in case of >/= 1,000,000 copies. As this, however, is the concentration at which even traditional immunochromatographic rapid tests (RDTs) start becoming reliable, it's quite difficult to see why such a molecular test should be performed at all? And not just an RDT? In particular if a pandemic shall be contained within its early stages, high sensitivity in spite of low copy numbers is usually desired to make sure that a test shows a positive result as early as possible in the course of the infection.

4. Discussion, first paragraph, last sentence: There have been numerous molecular SARS-CoV-2 assays producing results faster than within 1 hour – so what is so particular about this timeline?
